# The Protective Role of Self-Compassion in the Relationship between Perfectionism and Burnout in Portuguese Medicine and Dentistry Students

**DOI:** 10.3390/ijerph19052740

**Published:** 2022-02-26

**Authors:** Ana Telma Pereira, Maria João Brito, Carolina Cabaços, Mário Carneiro, Frederica Carvalho, Andreia Manão, Ana Araújo, Daniela Pereira, António Macedo

**Affiliations:** Institute of Psychological Medicine, Faculty of Medicine, University of Coimbra, Rua Larga, 3004-504 Coimbra, Portugal; emejotaaa@gmail.com (M.J.B.); csm.cabacos@gmail.com (C.C.); mario.carneiro@outlook.pt (M.C.); frederica.carv@gmail.com (F.C.); andreiamanao.r@gmail.com (A.M.); araujo.ana90@gmail.com (A.A.); dsmpereira4@gmail.com (D.P.); amacedo@ci.uc.pt (A.M.)

**Keywords:** perfectionism, self-compassion, burnout, medical students

## Abstract

Recent studies have documented the high prevalence of burnout among medicine and dentistry students, with potentially catastrophic consequences for both students and patients. Both environmental and personality factors play a part in burnout; perfectionism, a common trait in medicine students’ personalities, has been linked to psychological distress and increasing students’ vulnerability to burnout. Self-compassion, i.e., treating oneself kindly through hardship, has recently emerged as a buffer between perfectionism and psychological distress. While using a novel three-factor conceptualization of perfectionism (BIG3), this study aims to analyze if self-compassion has a protective role in the relationship between perfectionism and burnout, in a sample of medicine and dentistry students, through mediation analysis. We found that self-compassion significantly mediated the relationship between all three forms of perfectionism and burnout: as a partial mediator in self-critical and rigid perfectionism, as well as a full mediator in narcissistic perfectionism. Our findings underline self-compassion’s relevance in burnout prevention and management, supporting its use as an intervention target in burnout reduction programs and strategies.

## 1. Introduction

Burnout is a psychological syndrome that occurs following a prolonged period of intense stress, related to work and/or study, which results in a depletion of emotional and physical resources [1]. It is associated with motivational and performance decreases and other psychological difficulties [2]. The definition of burnout has emerged with Freudenberger [3] and includes three components: emotional exhaustion (which refers to feelings of emotional resources being depleted and represents the stress dimension of burnout), depersonalization (or cynicism) (which represents the interpersonal dimension of burnout), and feelings of ineffectiveness and lack of achievement (that represents the self-evaluation dimension of burnout).

It occurs at all stages of medical education and career [4] and is increasingly recognized as a public health and social problem, due to its high prevalence and negative impact on personal health, patient care, and economic costs [5]

The prevalence of burnout among medical [6] students is estimated to be higher than 40%, with no significant gender or medical education stage (pre-clinical vs. clinical) differences. Burnout symptoms and consequences increase from pre- to postgraduate medical education, reaching their peak in the residency [7].

The high rates of burnout are also observed in dentistry students [8,9], as well as its negative consequences [10].

Medical students with burnout have increased rates of depression, anxiety, substance dependence/abuse [11,12], and suicidality [13]. Students with burnout are three times more likely to have suicidal ideation [14], and the prevalence of suicidal ideation among medical students worldwide is higher than in any other course [15]. These are evidences that should alarm the medical education community worldwide. Additionally, in this context, medical students think more about dropping out of medical school and have poorer academic results [16]. 

It is important to keep in mind that the negative consequences of burnout are not just for students/doctors, but also for patients. Mental distress is associated with decreased patient safety and increased difficulties and errors in clinical decisions [17,18]. For example, students with high burnout are more likely to report normal physical findings, when the relevant physical exam is omitted [19].

Furthermore, as burnout increases, empathy (emotional component) and compassion (behavioral component) decrease [20]. It is recognized that these are elements of crucial importance in treatment success and patient satisfaction [21] and, as such, essential, not only for the quality, but also the efficiency of medical care [22].

Entering in a medical course, and pursuing the objectives to complete it, may represent a great achievement in a student’s life, but it may also embody the most stressful time they have had to face to date. Medical schools have a unique structure and highly competitive culture that make it an exceedingly challenging environment [15]. Medical students typically encounter stressful situations, including high workload, high volume of information to memorize, many evaluations and assessments, the pressure and chaos of clinical environment, numerous responsibilities, anxiety regarding their grades, long hours of studying, sleep deprivation [23], and, more recently, concerns about their future career. Young adults transitioning to medical school face other external sources of stress, such as the lack of time to engage in social and leisure activities, first-time exposure to sickness, pain, suffering, and death, the separation from family and friends, and/or the financial strains [24,25].

In the last two academic years, COVID-19 has brought new challenges to medical and dentistry education [26,27], such as the online learning environment, distance education, anxiety about their removal from clinical practice, and introduction of novel methods of student assessment, with potential implications on their academic and mental health trajectories. Other potential stressors were common to other population groups, such as the imposition of unfamiliar public health measures, including social distancing and lockdown, social fear related to COVID-19, fear of being infected by SARS-CoV-2, worry about older relatives, and the abrupt switch to a new reality [28]. This has been associated with high levels of psychological distress in the general population [29,30] and, more prominently, in health professionals [31] and college students [32].

Although a recent systematic review and meta-analysis indicated that the prevalence of anxiety in medical students is similar to that observed prior to the pandemic [33], a study carried out in Portugal reports evidence of the negative impact of COVID-19 on medical and dentistry students’ well-being, whose depression, anxiety, and stress scores were significantly lower before than during the pandemic, when the fear of COVID-19 was a significant predictor of psychological distress [34]. In this study, it was also found that the sources of stress, most associated with Portuguese medical students’ burnout during the pandemic, were, apart from those related to the high course demands and unhealthy lifestyle, the COVID-19 related restrictions imposed on training and socializing [30]. Other studies confirm the COVID-19 impact on medical [35,36] and dentistry [37,38] students’ burnout.

The environmental factors, already mentioned, contribute significantly to the development of burnout. However, we must not forget the importance of the complex interplay between these contextual and personal factors.

It has been proposed that the medical course itself may attract individuals with certain dispositional characteristics, such as high levels of perfectionism and conscientiousness [7], as well as the facets encompassed in the “triad of compulsiveness” (self-doubt, guilt, and exaggerated sense of responsibility [39]) [40].

Perfectionism has generally been conceptualized as a motivational trait [41] that can be more or less salient, depending on the situation [42,43]. The studies and academic domains are contexts where perfectionism is often reported to be the most salient [44,45], as task-specific performance assessments and achievement-related indicators are ubiquitous [46]. Definitions of perfectionism center on the pursuit of high standards, striving for flawlessness, setting excessively high standards for performance, and evaluating one’s own behavior overly critically [47]. Although some of the perfectionism facets may be valued and promoted as adaptive, in the sense that they may serve the best interest of individuals, other facets have been consistently found to be maladaptive, due to their association with high levels of psychological distress, namely in medical students. We have found that this trait leads to depression, anxiety, and fatigue by increasing levels of negative repetitive thinking (worry and rumination) [48], as well as of other negative cognitive emotional regulation strategies, such as catastrophizing [49].

Studies by several research groups, including our own, have shown that medical students have higher levels of perfectionism, compared to samples of the general population and even students from other fields [8,50,51]. Paradoxically, this may constitute another vicious cycle, as a recent review with meta-analysis confirms that perfectionistic concerns are negatively associated with academic achievement [52], which may reinforce perceptions of failure and not being or doing enough.

Two recent meta-analyses confirmed that perfectionism is positively related to burnout [2], occupational-related stress, anxiety, and depression [53].

In September 2020, Anne Helen Petersen, an American writer and former academic, published a book titled “Can’t Even”, born from an essay that went viral in 2019; its subtitle was: “How Millennials Became the Burnout Generation”. The author argues that burnout is a definitional condition for the generation born between 1980 and 2000. In this book, we can find many stories of young people in suffering and realize that they all share the trait of perfectionism.

A recent model for evaluating perfectionism distinguishes three broad dimensions, i.e., self-critical, rigid, and narcissistic perfectionism [54], have already been found to be associated with depression, anxiety, and stress in medicine and dentistry students [55], but not specifically with burnout.

Rigid perfectionism consists in demanding flawless performance from the self [54]; it contains the facets self-oriented perfectionism and self-worth contingencies, which reflect the personal strivings towards perfection, as well as the perception of one’s self-worth as being dependent on meeting personal standards of perfection [54]. Self-critical perfectionism is operationalized in the Dunkley Zuroff and Blankstein et al. conception [42], including concern over mistakes (overly negative reactions to perceived mistakes and failures), doubts about actions (pervading uncertainty and dissatisfaction of one’s performance), self-criticism (overly self-critical responses to perceived absence of perfection), and socially-prescribed perfectionism (the perception that others demand perfection from oneself). Narcissistic perfectionism is defined as a tendency to demand perfection from others in a grandiose, hypercritical, and entitled way [54]. 

Returning to empirical evidence, a recent, large meta-analysis showed significant increases in students’ levels of perfectionism, and, in their perceptions, that others are more demanding of them, others, and themselves than previously observed. This led experts to consider perfectionism a serious, even deadly, epidemic in modern western societies [56]. Thus, research has shown that our increasingly competitive society puts dysfunctional perfectionists at high risk of exhaustion and psychological distress, but also that self-compassion may be a potent antidote for this social “poison”.

Self-compassion is defined by a kind, warm, and caring attitude toward oneself, entailing a positive view of the self and the recognition that personal shortcomings are only human [57].

As self-compassion increases, both psychological distress and perfectionism decrease [58,59,60,61]. More specifically, after finishing their courses, young physicians with higher levels of burnout have lower mindfulness and self-compassion skills [62]. 

Self-compassion mediates the relationship between perfectionism and depression [63,64,65] and, in a recent study with psychology students, between self-critical perfectionism and burnout [66]. Intervention studies (RCTs) have already proved that a mindful self-compassion training decreases perceived and biological stress levels and maladaptive perfectionism [67]. The relationship between perfectionism, self-compassion, and burnout has not yet been studied in medicine and dentistry students.

The goal of this work is to analyze the role of perfectionism and self-compassion on overall burnout levels of medical and dentistry students. Our hypothesis is that their perfectionism will predict low self-compassion that will, in turn, reinforce their burnout; that is, self-compassion will mediate the relationship between perfectionism and burnout in this population group. If proven, this may substantiate and encourage the development, implementation, and validation of self-compassion-based programs, in order to reduce this problem, which directly affects our students and, indirectly, patients.

## 2. Materials and Methods

### 2.1. Procedure and Participants

Participants were recruited from January–February and June–July 2020, through social networks and using Google Forms.

The sample was composed of 528 participants, medical (80.3%; *n* = 424) and dentistry (19.7%, *n* = 104) students, the majority of which from the University of Coimbra (42.2%, *n* = 223) and University of Lisbon (27.3% *n* = 144). Little more than half of the students (54.2%, *n* = 286) were in their pre-clinical years (1st–3rd year).

Out of the total 528 students, 437 (82.8%) were female. Age ranged from 18 to 41, with a mean age of 21,34 (±29.17) years old. More than half of the participants were up to 21 years old (*n* = 305; 57.77%). The vast majority was Portuguese (94.9%; *n* = 501), and all students were fluent in the Portuguese language.

### 2.2. Measures

All the questionnaires used in the present study revealed good reliability and validity (construct and concurrent) in Portuguese samples. The internal consistency coefficients (Cronbach’s alpha), obtained with the sample of this study, are presented in Table 1.

#### 2.2.1. Perfectionism

The Big-3 Perfectionism Scale (BTPS) [54] is composed of 45 items that assess three higher-order global factors (rigid perfectionism, self-critical perfectionism, and narcissistic perfectionism) via ten lower-order perfectionism facets (self-oriented perfectionism, self-worth contingencies, concern over mistakes, doubts about actions, self-criticism, socially prescribed perfectionism, other-oriented perfectionism, hypercriticism, grandiosity, and entitlement). Additionally, the original version, the Portuguese BTPS, presented good validity (construct, concurrent, and convergent-divergent) and reliability for the underlying three-factors structure that overlapped with the original [55].

#### 2.2.2. Burnout

The Maslach Burnout Inventory-Student Survey (MBI-SS) is an adaptation of the Maslach Burnout Inventory, the most widely used instrument in assessing burnout. It is a self-report measure, comprising of 15 items that evaluate three factors: exhaustion, disengagement, and academic efficacy. Both the original version [68] and Portuguese translation [69] showed a good model fit to data and adequate construct validity and reliability for the three-factor structure of burnout. In another study (carried out with a larger sample that included participants of the present study), the second order factor, the overall burnout measure, presented a good fit [70]. Given this, the total score was used.

#### 2.2.3. Self-Compassion

The self-compassion scale (SCS) is the most used measure in the study of self-compassion in undergraduate samples. It is composed of 26 items, organized in three positive factors and three negative factors, part of three opposing pairs: self-kindness vs. self-judgment; common humanity (which consists of seeing the experience of our life, as part of the experience of humanity) vs. isolation; and mindfulness (which consists on holding a balanced awareness) vs. over-identification [71].

The Portuguese version has shown good model fit to data, as well as good internal consistency, factorial validity, and convergent validity [72], including a sample exclusively composed of medicine students [73].

### 2.3. Data Analysis

Descriptive, *t*-test, and Pearson correlation analyses were conducted using IBM^®^ SPSS^®^ Statistics, version 27. 

Correlations were used to investigate if the independent, dependent, and mediator variables were correlated to each other. The levels of correlation coefficients were defined as follows: 0.10—low, 0.30—moderate, and 0.50—high [74]. The mediation analyses were performed using PROCESS macro (Model 4) for SPSS^®^ [75]. Mediation models were tested to examine the mediation role of self-compassion in the relationship between perfectionism (self-critical/rigid/narcissistic) and burnout. Perfectionism was hypothesized to have a pernicious effect on compassion that, in turn, was supposed to reinforce burnout. 

The effects were estimated with 5000 bias-corrected bootstrap samples. The PROCESS macro uses the bootstrapping method, which is a method of assessing direct and indirect effects of variables, in a way that maximizes power and is robust against non-normality. The indirect effect represents the impact of the mediator variable(s) on the original relation (i.e., the relation of the independent variable on the outcome variable). If 0 is not contained within the confidence interval (CI), this suggests that the difference between the total and direct effects was different from 0 and, thus, that the indirect effect is significant. Typically, 95% bias-corrected (BC) bootstrap confidence intervals are used to judge the significance of the indirect effect, with confidence intervals resampled 5000 times for each analysis [75]. The three perfectionism dimensions were the independent variables, and the burnout total score was the dependent variable, resulting in three separate mediation models. 

## 3. Results

### 3.1. Descriptive and Correlation Analysis

Table 1 presents the descriptive data, internal consistencies obtained for each scale, and Pearson’s correlation coefficients between all variables.

The mean comparison of all the variables by gender revealed that women had significantly higher levels of self-critical perfectionism (57.94 ± 14.28 vs. 52.58 ± 12.78, *t* = −3.312, *p* = 0.001) and significantly lower levels of self-compassion (2.87 ± 0.70 vs. 3.05 ± 0.74, *t* = 2.196, *p* = 0.029). 

All the three perfectionism variables correlated with self-compassion and burnout, with coefficients of moderate to high magnitude for self-critical and rigid perfectionism and low magnitude narcissistic perfectionism (all *p* < 0.01). Thus, the proposed mediator, self-compassion, was correlated with the proposed predictors (perfectionism dimensions) and outcome variable (burnout).

### 3.2. Mediation Analyses

As there were significant differences between genders in the mean scores of self-critical perfectionism and self-compassion, gender was statistically controlled in Model 1 (entered as a covariate).

Table 2 presents the summary of the results of the mediation analysis, with an indication of the direct (c′) and indirect (c) effects that were estimated for all mediations.

The first model, presented in Figure 1, tested whether self-compassion would mediate the relationship between self-critical perfectionism and burnout (Figure 1).

Results indicated that the direct effect of self-critical perfectionism on burnout was significant (effect = 0.2608, SE = 0.0401, *t* = 6.5059, *p* < 0.001). Table 2 also shows that the indirect effect was 0.0532 and statistically different from zero (95% CI: 0.0029 to 0.1030). This model explained 18.42% of burnout variance (F = 39.43, *p* < 0.001).

The second model, presented in Figure 2, tested whether self-compassion would mediate the relationship between rigid perfectionism and burnout (Figure 2).

The direct effect of rigid perfectionism on burnout was significant (effect = 0.2435, SE = 0.0431, *t* = 4.5844, *p* < 0.001). The indirect effect was 0.1343 and statistically different from zero (95% CI: 0.0859 to 0.1868). This model explained 15.18% of burnout variance (F = 46.9667, *p* < 0.001). 

The third model, shown in Figure 3, tested whether self-compassion would mediate the relationship between narcissistic perfectionism and burnout (Figure 3).

The direct effect of narcissistic perfectionism on burnout was not significant (*p* = 0.0747). The indirect effect was 0.0654 and statistically different from zero (95% CI: 0.0332 to 0.0989) (Table 2). This model explained 12.31% of burnout variance (F = 36.8631, *p* < 0.001).

## 4. Discussion

To our knowledge, this is the first study designed to investigate the role of perfectionism on self-compassion and burnout levels in medical students. Our hypothesis that perfectionism predicts low self-compassion and increases the vulnerability to experience burnout during medical school was confirmed. 

As proposed by Hill and Curran [2], there is strong evidence on the relationship between perfectionism and burnout. However, few studies have focused on mediating factors, with coping being the most commonly examined construct. In the same meta-analysis, the authors suggest additional research in education is required, considering the highly relevant role of perfectionism and burnout in schools and universities [76]. Therefore, our work adds value to the existing literature by studying, in a sample of Portuguese medical students, the perfectionism-burnout relationship and mediating role of self-compassion, as a potential target for future interventions, aimed at reducing perfectionism driven burnout. 

Perfectionism is a personality trait of interest in the medical field, not only because medical and dentistry students have been shown to present higher perfectionism scores than their peers in other study fields, but also because it has been implicated in anxiety [77] depression [78], and burnout [8,79,80]. 

According to the diathesis-stress model [42,81], for students with high perfectionism, external and internal sources of anxiety and stress during medical school can be even more distressing, as the exceedingly high expectations for themselves and others are unlikely to be met. Depression, anxiety, and burnout symptoms or syndromes may arise, as well as substance abuse or even suicidal behavior [82]. 

In our study, all three perfectionism dimensions showed significant negative relation with self-compassion and positive relation with burnout, even the novel narcissistic perfectionism that includes Hewitt and Flett’s other-oriented perfectionism, which had been previously suggested to have deleterious effects, mainly on the relationships with others [83]. Perfectionism positively predicted burnout via low self-compassion. In other words, the latter acted as a mediator in that relationship, suggesting it may have a protective role in preventing burnout in highly perfectionist students. 

The application of a new instrument (BTPS) that encompasses evolution in social and cultural contexts in order to measure self-critical, rigid, and narcissistic perfectionism added comprehensive value to our results.

We began by analyzing gender differences in our variables’ distributions (perfectionism, self-compassion, and burnout), which revealed that women have significantly higher levels of self-critical perfectionism and lower self-compassion than men, with no statistically significant differences concerning burnout levels. Our results corroborate previous data, using the same instruments, concerning gender differences in perfectionism [34] and self-compassion [84]. Additionally, previous studies had already identified higher general stress levels in female students, without significant gender differences in burnout levels, assessed with MBI [85]. 

What is innovative in this study, and constitutes its main strength, is the evidence that high perfectionism increases vulnerability to burnout in medical students, through the effect of low self-compassion. 

Regarding the three second-order perfectionism dimensions, self-critical perfectionism presented correlations of higher magnitude with self-compassion and burnout, followed by rigid perfectionism and, at last, narcissistic perfectionism. In other studies, among the three dimensions, self-critical perfectionism was the one that was most strongly associated to poorer psychological outcomes, such as depression, anxiety, and stress [86].

Self-critical perfectionism predicted burnout, both directly and indirectly, via the effect of low self-compassion. This means that perfectionism leads to burnout, independently of self-compassion, though one of its pathways of influencing this outcome may be through self-compassion.

Rigid perfectionism increases vulnerability to burnout, both directly and indirectly, through the mediating effect of low self-compassion. 

Self-critical perfectionism and rigid perfectionism globally refer to evaluative concerns and excessively high personal standards, respectively. In medical students, these dimensions reflect the “culture of perfection”, which reinforces certain traits and dysfunctional beliefs, for which they are already prone to and strongly predict medical students’ distress and burnout [87]. Thus, the influence of each of these perfectionism dimensions on burnout levels does not depend on the concomitant effect of low self-compassion, although the latter reinforces that effect, acting as a partial mediator. Our results add to Hill and Curran’s [2] meta-analysis on multidimensional perfectionism and burnout by using a new measure of multidimensional perfectionism. The meta-analysis focused on studies examining perfectionistic strivings and perfectionistic concerns, which are globally included in BTPS’ self-critical and rigid perfectionism, respectively, the latter showing stronger relationships with burnout and burnout symptoms, as expected. By using BTPS’ three global perfectionism factors, we were able to meet a novel multidimensional conceptualization of perfectionism, with the unique opportunity to capture narcissistic perfectionism.

The correlations between narcissistic perfectionism, i.e., the belief that one is superior to others and should be treated as such [88], and burnout showed a lower magnitude, compared to other perfectionism dimensions in our study, as well as to what we would expect using the trait narcissism, based on previous literature [89,90,91], indicating a less pernicious nature of the narcissistic type of perfectionism, which only becomes apparent when the demands from the environment are too stressful and prolonged over time, thus jeopardizing the individual’s subjective ability to be perfect. 

When considering this relationship between narcissistic perfectionism and burnout, another concept comes to mind: what about self-esteem? Traditionally lauded by western culture as practically synonymous with psychological well-being, in more recent years, the alleged benefits of high self-esteem have been under questioning. When looking at the early conceptualization of self-esteem, i.e., the degree to which the self is judged to be competent in life domains deemed important [92], we could infer that narcissistic individuals should have high self-esteem levels; shouldn’t this protect them from psychological distress? 

Important to narcissistic individuals is the visibility of their accomplishments and being acknowledged [93]. There is often a discrepancy between self-awareness and the perception of others, so that narcissistic individuals experience feelings of being misunderstood and offended. As their self-esteem is unstable and highly contingent on particular outcomes and the perceived opinions of others [94], narcissists may work too hard and do too much, while neglecting private life, in order to pursue perfection. By doing so, they might be trying to protect their fragile sense of self from the shame that would arise from the loss of admiration from others, if any mistakes or insufficiencies were to be found in their work [95].

More than self-esteem, our findings, i.e., that narcissistic perfectionism does not directly lead to burnout, requiring full mediation by low self-compassion, highlight the importance of self-compassion, revealing that these perfectionistic and grandiose self-expectations and the tendency to be hypercritical toward others are not sufficient on their own for predicting burnout in medical students. This means their negative influence is only noticeable when self-judgement, isolation, and over-identification of thoughts and feelings are also present. 

As our findings support, self-compassion can be one of the answers to the high prevalence of burnout among undergraduates.

A person with high self-compassion will be able to see their own shortcomings accurately and react to them in a lenient and equanimous manner, recognizing that their inadequacies are a part of the shared human experience [96]. Several studies have demonstrated the importance of self-compassion in burnout levels [97,98,99], with some focusing on its role in buffering the pernicious influence of self-critical perfectionism on psychological well-being [66,100]. It is fairly intuitive to understand how self-compassion might help perfectionists: by replacing the harsh self-judgements that often accompany perfectionistic evaluative concerns with tolerance and kindness toward one’s imperfections [64], self-compassion might ease the cognitive and emotional burden of perfectionism.

The results of this work provide further encouragement for the implementation of burnout prevention and management programs, based on self-compassion training, especially in perfectionists. Although several studies have already examined the impact of different self-compassion-based interventions on highly self-critical individuals [101,102,103], it would be interesting to assess the effect of these strategies on other types of perfectionism.

Our findings should be considered in light of the limitations of the study. The most important limitation is the cross-sectional study design. Longitudinal studies would be necessary to test mediation appropriately and help establish causality. The high proportion of female participants alerts us to be cautious with gender-based generalizations. Notwithstanding, the effect of gender was controlled as a covariate in the mediation model, where it could act as a confounding variable. Furthermore, this differential proportion is representative of this Portuguese populational group, as medicine and dentistry courses are much more frequented by girls (about five times more) than by boys. Another possible limitation is that students with higher levels of perfectionism and/or burnout may have been more motivated to participate.

Examining the same relations and pathways on medical residents may be a promising endeavor for future research. 

## 5. Conclusions

This study highlighted, for the first time, the role of perfectionism on medical students’ burnout, using a novel multidimensional conceptualization of perfectionism. Perfectionism negatively influences medical students’ wellbeing, predisposing them to burnout, and this effect is mediated by low self-compassion. Academics with high levels of perfectionism should be helped in dealing with their dysfunctional cognitions and beliefs, namely through interventions focused on enhancing self-compassion.

## Figures and Tables

**Figure 1 ijerph-19-02740-f001:**
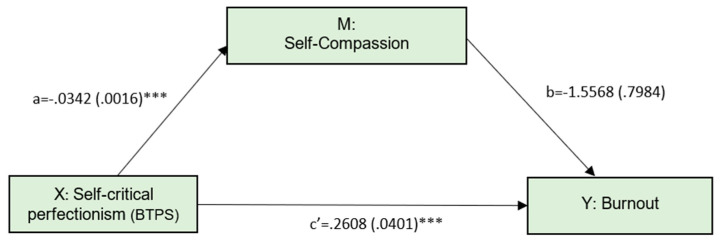
Simple mediation model, with self-critical perfectionism as the predictor. Gender was controlled and did not present a significant effect. Numbers represent unstandardized coefficients. Numbers in parentheses represent standard errors. *** *p* < 0.001.

**Figure 2 ijerph-19-02740-f002:**
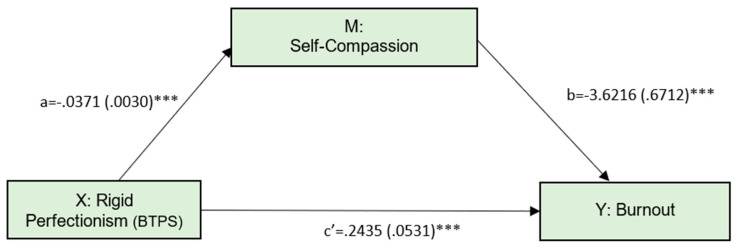
Simple mediation model, with rigid perfectionism as the predictor. Numbers represent unstandardized coefficients. Numbers in parentheses represent standard errors. *** *p* < 0.001.

**Figure 3 ijerph-19-02740-f003:**
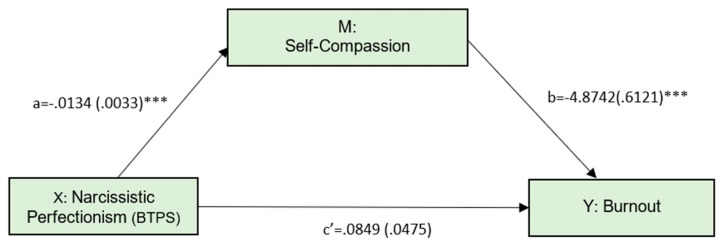
Simple mediation model, with narcissistic perfectionism as the predictor. Numbers represent unstandardized coefficients. Numbers in parentheses represent standard errors. *** *p* < 0.001.

**Table 1 ijerph-19-02740-t001:** Descriptive statistics, internal consistencies, and correlations.

	2	3	4	5	Mean	SD	α
1 SCritPerf	0.75 **	0.31 **	−0.68 **	0.42 **	570.02	140.17	0.86
2 RigPerf	1	0.46 **	−0.47 **	0.32 **	270.10	80.93	0.85
3 NarcPerf		1	−0.17 **	0.13 **	300.88	90.11	0.85
4 Self-Comp			1	−0.34 **	20.90	0.71	0.73
5 Burnout				1	400.68	100.43	0.73

Note: SCritPerf: self-critical perfectionism; RigPerf: rigid perfectionism; NarcPerf: narcissistic perfectionism; SD: standard deviation; ** *p* < 0.01.

**Table 2 ijerph-19-02740-t002:** Direct and indirect effects of the mediation models, with burnout as the outcome.

					Bootstrapping BC 95% CI	
MODEL		Coefficient	SE	*p*	Lower	Upper
1	PREDICTOR: Self-critical perfectionism					
Covariate: Gender	Direct effect c′	0.2608	0.0401	<0.001	0.1820	3395
	Indirect effect c	0.0532	0.0255		0.0029	0.1030
2	PREDICTOR: Rigid perfectionism					
	Direct effect c′	0.2435	0.0531	<0.001	0.1392	0.3479
	Indirect effect c	0.1343	0.0258		0.0859	0.1868
3	PREDICTOR: Narcissistic perfectionism					
	Direct effect c′	0.0849	0.0475	0.0747	−0.0085	0.1782
	Indirect effect c	0.0654	0.0168		0.0332	0.0989

Note: SE: Standard error.

## Data Availability

The data presented in this study can be available on request from the corresponding author. The data are not publicly available, due to ethical and privacy restrictions.

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
