# Peer review of "The Protective Role of Self-Compassion in the Relationship between Perfectionism and Burnout in Portuguese Medicine and Dentistry Students"

_ijerph, 2022, doi:10.3390/ijerph19052740_

Round 1

Reviewer 1 Report

Comments and suggestions for authors are attached in a word document

Author Response

Thank you very much for your positive comments and for the opportunity of improving the manuscript. We have appreciated the comments and have accepted the suggestions. Please find below our answers.

  1. A s suggested, the three components of the perfectionism trait (self-critical perfectionism, rigid perfectionism and narcissistic perfectionism) are now described in the introduction, more specifically in the first part of the theoretical foundation.

  1. We have added a note recommending caution in generalizations based on gender, given the much larger number of female participants. However, we have also pointed out that this differential proportion is representative of this Portuguese populational group, as medicine and dentistry courses are much more frequented by girls (about five times more) than by boys.

Reviewer 2 Report

1. Thank you for the opportunity to review this paper. The topic seems to be important and relevant. However, the authors does not succeed to clearly argue for why it is relevant to investigate the the role of perfectionism and self-compassion on overall burnout levels of medical and dentistry students. This could be improved to in order to justify the importance of the topic under consideration.

2. The introduction would benefit from being restructured and condensized so that the reader easily can follow what the problem is, what we already know on the topic, what we do not know and how this study would add to this research gap.

3. The hypothesis is described in the introduction as: Our hypothesis is that self-compassion will mediate the relationship between that trait and that condition perniciously affecting this population group. Please build up the argument for the proposed hypothesis, i.e. so that it could be understod why it is a valid hypothesis.

4. Later, the hypotheses is expressed in a more precise way (line 245-256): "Perfectionism was hypothesized to have a pernicious effect on compassion, that in turn was supposed to reinforce burnout". At the beginning of the discussion the hypothesis is instead expressed in the following way (line 283-284): ”Our hypothesis that perfectionism predicts low self-compassion and increases vulnerability to experience burnout during medical school was confirmed.” It would be easier for the reader if the hypothesis is expressed in the same way throughout the paper. This is especially important when interpreting the findings of the study.

5. Please add an explanaition of the different types of perfectionisms that is used in this study and how these are expected to affect burnout.

6. How well does the reqruited sample match (or mirror) the study population in medicine or dentist school? Is it possible that different types of students are more prone to participate in this type of study than others? How might/will this affect your result?

7. The statistical section would benefit from including a description of the different steps in the analysis in addition to the information about the different tests used. For example, how was the different steps in the analyses conducted to investigate if the principles for mediation was fullfilled? Were the correlations used to investigate if the independent variables, the dependent variable as well as the mediator variable were correlated to each other in a first step? What levels of correlation coefficients were defined as low, moderate or high magnitudes? Please also add a reference to support the chosen definition of levels.

8. Parts of the description of the statistical analyses is described in the result section. This is better suited in the statistical section.

9. In figure 3 the direct relationship between narcissistic perfectionism on burnout was shown to be non-significant. I might have missunderstood something in the description of the statistical analyses procedure or the presentation of the result, but I interpret that it means that the basic requirements for mediation is not fullfilled since perfectionism is not significantly related to burnout (ref MacKinnon*). This could be expected since the correlation coefficient magnitude presented between narcissistic perfectionism and burnout (0.13) as well as narcissistic perfectionism and self-compassion (-0.17) in table 1 could be considered as low. Despite that the authors write that their hypothesis was confirmed. It might be that this intepretation needs to be adjusted and discussed in relation to the different parts of perfectionisms that were used and analysed.

*MacKinnon, D. P., Fairchild, A. J., & Fritz, M. S. (2007). Mediation analysis. Annual review of psychology, 58, 593–614. https://doi.org/10.1146/annurev.psych.58.110405.085542

10. Please double check to make sure that only results related to the aim of study are discussed in the discussion section.

11. The definitions of the different types of perfectionism should be introduced earlier in the paper rather than in the discussion, for example in the introduction section.

12. A section of potential limitations or methodological considerations should be added to the paper discussing weaknesses of, for example, the study design and potential effects on the result.

13. Please check to make sure that the aim in the abstract is consistens with how it is phrased in the other part of the manuscript.

Author Response

Thank you very much for the pertinent and useful revision and for the opportunity of improving our paper. We have appreciated the comments and have accepted all the suggestions. Please find below our answers and list of changes point by point, in accordance with your comments.

  1. In order to justify the importance of the topic under consideration, we have tried to improve the clarity of arguments about the relevance of investigating the role of perfectionism and self-compassion on overall burnout levels of medical and dentistry students.
  2. By clarifying those arguments (as suggested in point 1), the introduction is now somewhat condensed, so that the reader can better follow what the problem is, what we already know on the topic, what we do not know and how this study would add to this research gap.
  3. and 4. The hypothesis in the introduction is now rephrased, as follows:

”Our hypothesis is that medicine students’ perfectionism will predict low self-compassion that will in turn reinforce their burnout, that is, self-compassion will mediate the relationship between perfectionism and burnout in this population group”.

Furthermore, the hypothesis is now expressed in the same way throughout the paper, to facilitate the reading and interpretation of the study results.

  1. Also following a suggestion from another reviewer, we have added an explanation of the different types of perfectionism that were used in this study in the introduction; how these were expected to affect burnout was added to this revised version.
  2. We have added a sentence, in the discussion’s paragraph relative to the potential limitations, about the possibility that this bias (some students may be more prone to participate in this type of study than others) occurring.
  3. The statistical section has been edited to include a description of the different steps in the analysis, in addition to the information about the different tests used. For example, in the revised version it is now clarified that, in a first step, the Pearson correlations were used to investigate if the independent variables (or the predictors – perfectionism traits), the dependent variable (or the outcome – burnout) as well as the mediator variable (self-compassion) were correlated to each other;

the levels of correlation coefficients are now defined: .10 as low, .30 as moderate and .50 as high; a reference to support these levels was added - Cohen, J. (1977). Statistical power analysis for the behavioural sciences (rev. ed.) Lawrence Erlbaum Associates. Inc., Hillsdale, NJ, England

  1. Parts of the description of the statistical analyses that were described in the result section are now included in the statistical section.
  2. In the mediation analyses we have followed the Hayes (2013) approach, according to which when the direct effect is non-significant and the indirect effect is significant this means full or complete mediation, that is, the entire effect of an independent variable (in this model, represented in Figure 3, narcissistic perfectionism) on a dependent variable (burnout) is transmitted through the mediator variable (self-compassion). The requirement for mediation is that the correlation coefficients are significant, which was the case; although the Pearson coefficient’s magnitude between narcissistic perfectionism and burnout was low (0.13) as well as between narcissistic perfectionism and self-compassion (-0.17), they were significant. Given this, this result does not invalidate that our hypothesis was confirmed.
  3. Discussion section were double checked to make sure that only results related to the aim of study were discussed.
  4. As already mentioned, the definitions of the different types of perfectionism are now introduced earlier in the paper (in the introduction section).
  5. The penultimate paragraph of the discussion is precisely about the potential limitations of the study, namely its cross-sectional design, which limits conclusions on causality. The potential limitation concerning the high proportion of female participants was reflected in more depth.
  6. All the manuscript was check and edited when necessary to make sure that the aim in the abstract is consistent with how it is phrased in other parts of the manuscript.